# Dietary Taurine Improves Growth Performance and Intestine Health via the GSH/GSSG Antioxidant System and Nrf2/ARE Signaling Pathway in Weaned Piglets

**DOI:** 10.3390/antiox12101852

**Published:** 2023-10-12

**Authors:** Lingang Wang, Liwen Jiang, Yunyun Chu, Fu Feng, Wenjie Tang, Chen Chen, Yibin Qiu, Zhijin Hu, Hui Diao, Zhiru Tang

**Affiliations:** 1Laboratory for Bio-Feed and Molecular Nutrition, College of Animal Science and Technology, Southwest University, Chongqing 400715, China; wlg18700217533@163.com (L.W.); jlw18080401887@163.com (L.J.); 15234885526@163.com (Y.C.); ff_444444@163.com (F.F.); m15587125971@163.com (C.C.); 18223233084@163.com (Y.Q.); hzj-039@163.com (Z.H.); 2Animal Breeding and Genetics Key Laboratory of Sichuan Province, Sichuan Animal Science Academy, Chengdu 610066, China; wenhan28@126.com (W.T.); diao_hui@hotmail.com (H.D.); 3Livestock and Poultry Biological Products Key Laboratory of Sichuan Province, Sichuan Animtche Group Co., Ltd., Chengdu 610066, China

**Keywords:** taurine, weaned piglets, antioxidant capacity, intestine healthy, microorganism, growth performance, Nrf2/ARE signaling pathway

## Abstract

Early weaning of piglets was prone to increase reactive oxygen species, disrupt the redox balance, decrease antioxidant capacity, cause oxidative stress and intestinal oxidative damage, and lead to diarrhea in piglets. This research aimed to study dietary taurine (Tau) supplementation at a level relieving intestinal oxidative damage in early-weaned piglets. A total of 48 piglets were assigned to four groups of 12 individuals and fed a basal diet with 0.0% Tau (CON), 0.2% Tau (L-Tau), 0.3% Tau (M-Tau), or 0.4% Tau (H-Tau), respectively. The animal experiment lasted 30 days. The final weight, weight gain, average daily gain, and feed conversion rate increased with the increase in dietary Tau (Linear, *p* < 0.05; Quadratic *p* < 0.05), while the diarrhea index of piglets decreased with the increase in dietary Tau (Linear, *p* < 0.05). Serum malondialdehyde, nitric oxide (NO), D-lactose, and oxidized glutathione (GSSG) concentrations decreased with the increase in dietary Tau (Linear, *p* < 0.05). The O_2_^•−^ and ^•^OH clearance rate in serum, liver, and jejunum mucosa increased with the increase in dietary Tau (Linear, *p* < 0.05). Serum superoxide dismutase (SOD) activity, glutathione peroxidase (GPX) activity, catalase (CAT) activity, and peroxidase (POD) activity and total antioxidant capacity increased with the increase in dietary Tau (Linear, *p* < 0.05). The serum glutathione (GSH) concentration and the ratio of GSH to GSSG increased with the increase in dietary Tau (Linear, *p* < 0.05). The POD and glutathione synthase activity in the liver and jejunum mucosa increased with the increase in dietary Tau (Linear, *p* < 0.05). The mRNA abundances of *HO-1* and *GPX1* in the H-Tau group were higher than that in the L-Tau, M-Tau, and CON groups (*p* < 0.05). The mRNA abundances of *SOD1* and *Nrf2* in the M-Tau and H-Tau groups were higher than in the L-Tau and CON groups (*p* < 0.05). The mRNA abundance of *SOD2* in the L-Tau, M-Tau, and H-Tau groups was higher than in the CON group (*p* < 0.05). The VH and the ratio of VH to CD of jejunum and ileum increased with the increase in dietary Tau (Linear, *p* < 0.05). The mRNA abundances of *occludens 1* and *claudin 1* in the H-Tau group were higher than that in the CON, L-Tau, and M-Tau (*p* < 0.05). The mRNA abundance of *occludin* in the L-Tau, M-Tau, and H-Tau groups was higher than that in CON (*p* < 0.05). The abundance of Firmicutes increased with the increase in dietary Tau (Linear, *p* < 0.05), while Proteobacteria and Spirochaetota decreased with the increase in dietary Tau (Linear, *p* < 0.05). Collectively, dietary supplementation of 0.3% and 0.4% Tau in feed could significantly improve the growth performance and enhance the antioxidant capacity of piglets.

## 1. Introduction

In order to improve the reproductive rate of sows and the survival rate of piglets, early weaning isolation technology is commonly adopted to reduce the weaning time from 2 months to 21–28 days [1]. Piglets have to undergo nutritional, psychological, and environmental pressure [2,3], subsequently generating weaning stress [4]. An excessive concentration of oxygen species (ROS) is produced during weaning stress [5], and this oxidizes and damages the intestinal morphological structure and mucosal barrier of piglets. The damaged intestinal morphological structure reduces the digestion and absorption ability of food. Viruses and bacteria can invade and colonize the intestine, causing intestinal inflammation, microbiota imbalance, dyspepsia, diarrhea, and poor growth [6]. 

Taurine (Tau) is a semi-essential *β*-amino acid and the most abundant free amino acid in animal tissues [7], accounting for 25%, 50%, 53%, and 19% of the total free amino acids in liver, kidney, muscle, and brain tissues, respectively [8]. The significance of a high Tau concentration is not only to provide raw materials for protein synthesis (energy for body metabolism) but also to play its rich biological functions as antioxidant functions [9,10]. Tau can avoid the mistranslation of proteins related to the electron transport chain by strengthening the precise positioning of codons and anti-codons and maintaining the structural and functional integrity of the electron transport chain more efficiently, thus avoiding the leakage of electrons during the transport process and reducing the generation of reactive oxygen species (ROS) [11,12,13,14]. Tau can act on the Keap1-Nrf2/ARE antioxidant signaling pathway, promote NF-E2-related factor 2 (Nrf2) to enter the nucleus and bind to the small Maf protein (Maf) in the nucleus into a heterodimer, and act on the antioxidant reaction element (ARE) on the nucleic acid sequence to induce the expression of antioxidase genes [15]. Acting as a pH regulator in the glutathione redox system [16], exogenous supplementation of Tau can also increase the amount of glutathione synthesis in the body [17], thereby improving the antioxidant capacity of the body.

This study aimed to explore whether dietary taurine supplementation improves growth performance and intestine health via the GSH/GSSG antioxidant system and the NRF2/ARE signaling pathway in weaned piglets.

## 2. Materials and Methods

### 2.1. Experimental Material

Tau (A3820, 99.0%), pharmaceutical grade, was purchased from Shanghai Yuanju Biotechnology Co., Ltd., Shanghai, China. 

Piglets (6.71 ± 0.73 kg, Duroc × Landrace × Yorkshire) were weaned at 28 d and purchased from Chongqing Nongdianshan Agricultural Science and Technology Development Co., Ltd., Chongqing, China.

All experimental procedures involving piglets were approved by the License of Experimental Animals (IACAU-20211007-06) of the Animal Experimentation Ethics Committee of Southwest University, Chongqing, China.

### 2.2. Experimental Design and Diets

A total of 48 weaned piglets were assigned to 4 groups of 12 individuals and fed a basal diet with 0.0% Tau (CON), 0.2% Tau (L-Tau), 0.3% Tau (M-Tau), and 0.4% Tau (H-Tau), respectively. The basal diet was designed according to the nutritional requirements of NRC (2012) [18], and the ingredients and nutrient levels of the diet are shown in Table 1. 

The piglets were kept in a temperature-controlled room (28 ± 1.2 °C) and individually in pens (2.0 m length × 0.8 m width × 1.0 m depth). The feed was pre-fed for 3 days, and the formal experiment lasted for 30 days. Weaned piglets were fed and hydrated on an ad libitum basis. Feed intake was recorded every day. The diarrhea of piglets was scored according to diarrhea score criteria [19].

### 2.3. Sample Collection 

Four piglets were selected from each group on day 31 for the collection of blood and tissue samples. A 20 mL blood sample was collected from each piglet. The blood samples were left undisturbed for 30 min and then centrifuged at 4000× *g* for 10 min at 4 °C to harvest the serum samples for biochemical analysis and ELISA. 

After blood collection, piglets were anesthetized by intravenous injection of 50 mg/kg body weight pentobarbital sodium and slaughtered. Liver tissues were collected for biochemical analysis and ELISA. About 2 cm intestinal tissues from the middle segment of the jejunum and the middle segment of the ileum were collected in a 50 mL centrifuge tube filled with a fixed solution (100 mL of 40% formaldehyde, 4.0 g of acidic sodium phosphate, 6.5 g of disodium hydrogen phosphate, and 900 mL of distilled water) for the morphology of the jejunum and the ileum. Intestinal mucosa on the anterior, middle, and posterior segments of the jejunum and the ileum were rinsed with cold saline, then they were scraped gently with a scalpel blade and immediately frozen in liquid N_2_ for quantitative PCR analysis. Approximately 10 mL caecum contents were collected for analysis of 16S rDNA gene sequencing.

### 2.4. Biochemical Parameters of Oxidative Stress and Indicators of Intestinal Barrier Integrity

Serum superoxide dismutase (SOD) activity, glutathione peroxidase (GPX) activity, catalase (CAT) activity, peroxidase (POD) activity, total antioxidant capacity (T-AOC), glutathione (GSH), oxidized glutathione (GSSG), malondialdehyde (MDA), nitric oxide (NO) and D-lactose (D-LA) concentrations, and POD activity and GSHS activity of jejunum mucosa were determined according to the introduction of a commercial biochemical assay kit (Nanjing Jiancheng Bioengineering Institute, Nanjing, China). 

^•^OH was formed by a Fenton reaction between H_2_O_2_ and Fe^2+^. ^•^OH reacted with C_7_H_6_O_3_, and the product showed special absorption at 510 nm. By adding different samples to the system with the same concentration of ^•^OH, after 20 min of reaction, the active substance in the sample could remove part of ^•^OH and then reduce the absorption value at 510 nm; the ^•^OH clearance rate was calculated using the following formula: ^•^OH clearance (%) = [A blank − (A determination − A control)] ÷ A blank × 100%”. The detailed instructions for ^•^OH clearance rate in serum and liver were used to determine according to the introduction of a hydroxyl free radical assay kit (Nanjing Jiancheng Bioengineering Institute, Nanjing, China). 

O_2_^•−^ reacts with the reduced material to produce a purplish red compound, which has a characteristic absorption peak at 570 nm, and the removal ability of the sample with respect to O_2_^•−^ is negatively correlated with the absorption value at 570 nm. The O_2_^•−^ clearance rate was calculated using the following formula: I% = [1 − (A determination − A control) ÷ A blank] × 100%. The O_2_^•−^ clearance rate in the serum and liver was used to determine, according to the introduction of inhibition, and produce a superoxide anion assay kit (Nanjing Jiancheng Bioengineering Institute, Nanjing, China). 

The concentrations of occludin (OCLN), claudin-1 (CLDN-1), and zonula occludens 1 (ZO-1) of jejunum mucosa were determined by an ELISA kit (Quanzhou Risin Biotechnology Co., Ltd., Quanzhou, China).

### 2.5. Real-Time RT-PCR

The total RNA was extracted using a universal RNA lifting kit (Accurate Biology, AG21022), and then the concentration and purity of the total RNA (A_260/280_, A_260/230_) were detected using a nucleic acid detector. Samples with qualified concentration and purity were reverse-transcribed into cDNA using a reverse transcription kit (Accurate Biology, AG11728), followed by PCR analysis using a fluorescent quantitative kit (Accurate Biology, AG11701) using the SYBR Green method. The mRNA abundances of *Nrf2*, heme oxygenase-1 (*HO-1*), *SOD1*, *SOD2*, *GPX1*, and Glutathione-S-Transferase (*GST*) were measured in liver samples with *GAPDH* as the internal reference gene. The mRNA abundances of *ZO-1*, *OCLN*, and *CLDN-1* were determined by jejunum mucosal samples. The obtained Ct values were normalized by the 2^−ΔΔCt^ method. The gene primer sequences of internal reference genes and target genes obtained from NCBI are shown in Table 2.

### 2.6. H and E Staining

The morphology of the jejunum and the ileum was analyzed according to the H and E staining method. An optical microscope (Carl Zeiss Inc., Oberkochen, Bayern, Germany) was used to view the sliced sample, and digital images were taken using a color video camera Sony 3CCD-VX3 camcorder (Sony Group Corp., Tokyo, Japan). The villus height and crypt depth were obtained using image analysis software cellSens Standard 1.12 (Olympus, Tokyo, Japan).

### 2.7. Cecal Microbial 16S rDNA Sequencing Analysis

The cecal contents were collected with aseptic centrifuge tubes and were sent to frozen storage at −80 °C, and then they were sent to Hangzhou Lianchuan Biotechnology Co., Ltd., Hangzhou, China, for microbial isolation of cecal contents. Then, 16S rDNA high-throughput sequencing was used to analyze cecal microbiota. 

### 2.8. Data Treatments and Statistic Analysis

The average daily weight gain (ADG), average daily feed intake (ADFI), and feed conversion rate (FCR) were calculated according to the following formulas: Average daily gain (ADG) = (final weight − initial weight)/days; Average daily feed intake (ADFI) = total feed intake/days; Feed conversion rate = average daily gain/average daily feed intake.

All data were presented as means ± Standard Error of Mean (SEM) and subjected to one-way analysis of variance using the general linear model procedure in the SAS 9.4 statistical software (SAS Institute, Inc., Cary, NC, USA). The LSD test was performed to identify differences between groups. Significance was set at *p* < 0.05.

## 3. Results

### 3.1. Growth Performance

As shown in Table 3, the final weight, weight gain, ADG, and feed conversion rate increased with the increase in dietary Tau (Linear, *p* < 0.05; Quadratic *p* < 0.05), while the diarrhea index of the piglets decreased with the increase in dietary Tau (Linear, *p* < 0.05). 

### 3.2. Antioxygenic Capacity

As shown in Table 4, the serum MDA, NO, and GSSG concentration decreased with the increase in dietary Tau (Linear, *p* < 0.05). The O_2_^•−^ and ^•^OH clearance rate in the serum, liver, and jejunum mucosa increased with the increase in dietary Tau (Linear, *p* < 0.05). Serum SOD activity, GSH-PX activity, CAT activity, POD activity, and T-AOC increased with the increase in dietary Tau (Linear, *p* < 0.05). The serum GSH concentration and the ratio of GSH to GSSG increased with the increase in dietary Tau (Linear, *p* < 0.05).

The POD and GSHS activity in the liver and jejunum mucosa increased with the increase in dietary Tau (Linear, *p* < 0.05).

As shown in Figure 1, the mRNA abundances of *HO-1* and *GPX1* in the H-Tau group were higher than that in the L-Tau, M-Tau, and CON groups (*p* < 0.05). The mRNA abundances of *SOD1* and *Nrf2* in the M-Tau and H-Tau groups were higher than in the L-Tau and CON groups (*p* < 0.05). The mRNA abundances of *SOD2* in the L-Tau, M-Tau, and H-Tau groups were higher than in the CON group (*p* < 0.05). 

### 3.3. Intestinal Morphology

As shown in Table 5, the VH and the ratio of VH to CD of jejunum and ileum increased with the increase in dietary Tau (Linear, *p* < 0.05), while the length, length index, weight, and weight index of jejunum and ileum had no significant changes (*p* > 0.05). As shown in Figure 2, compared with the CON group, the H-Tau and M-Tau groups had sharper inter-villous boundaries, fewer damaged microvilli in the villus and intervillus space, and a more orderly and consistent arrangement of microvilli.

### 3.4. Intestinal Mucosal Barrier

As shown in Table 6, serum D-LA content decreased with the increase in dietary Tau (Linear, *p* < 0.05), while the protein concentrations of CLDN-1, ZO-1, and OCLN in jejunum mucosa increased with the increase in dietary Tau (Linear and Quadratic, *p* < 0.05). 

As shown in Figure 3, the mRNA abundance of *ZO-1* and *CLDN-1* in the H-Tau group was higher than that in the CON, L-Tau, and M-Tau (*p* < 0.05). The mRNA abundance of OCLN in the L-Tau, M-Tau, and H-Tau groups was higher than that in the CON (*p* < 0.05).

### 3.5. Cecum Microbiota

As shown in Table 7, compared with the CON group, there were no significant differences in the observed_otus, shannon, simpson Chao1, pielou_e, and goods_coverage indexes of the H-Tau, M-Tau, and L-Tau groups (*p* > 0.05). Compared with the CON group, with the increase in the Tau supplemental dose, the abundance of Firmicutes increased (Linear, *p* < 0.05), while the abundance of Proteobacteria and Spirochaetota decreased (Linear, *p* < 0.05). Compared with the CON group, the abundance of *Clostridium_sensu_stricto_1* and *Turicibacter* in the H-Tau, M-Tau, and L-Tau groups were up-regulated with increasing Tau supplemental doses (Linear, *p* < 0.05). The abundance of the *Terrisporter* genus was up-regulated (*p* < 0.05), while the abundance of UCG-005 was down-regulated (Linear, *p* < 0.05).

As shown in Figure 4, 781 ASVs are unique to the CON group, 569 ASVs are unique to the L-Tau group, 575 ASVs are unique to the M-Tau group, and 441 ASVs are unique to the H-Tau group. The CON, L-Tau, and M-Tau groups share the 78 ASV species, and the CON, L-Tau, and H-Tau groups share the 59 ASV species, and the CON, M-Tau, and H-Tau groups share the 59 ASV species, and the L-Tau, M-Tau, and H-Tau groups share 183 ASV species. The CON group had the largest number of unique ASV species, and the L-Tau, M-Tau, and H-Tau combinations had the largest number of unique ASV species. 

As shown in Figure 5, at the phylum level, the branch distances of the L-Tau, M-Tau, and H-Tau groups and the CON group are greater than 0.06 units, while the branch distances of the L-Tau, M-Tau, and H-Tau groups are less than 0.04 units. Moreover, the branch distance between the Tau group and the CON group was positively correlated with the Tau supplemental dose. As shown in Figure 6, the distance between the L-Tau, M-Tau, and H-Tau groups was relatively close, and there were many overlapping parts.

## 4. Discussion

After the early weaning of the piglets, the activity of anti-oxidase dropped sharply, resulting in increased concentrations of the ROS and reactive nitrogen species (RNS), thus damaging proteins and lipids and causing oxidative damage to tissues and organs [20,21,22]. Therefore, the early-weaned piglets need antioxidant nutrient regulation. The results of this study showed that serum MDA, NO, and GSSG concentration significantly decreases with the increase in dietary Tau, and the O_2_^•−^ and ^•^OH clearance rates in the serum, liver, and jejunum mucosa significantly increased with the increase in dietary Tau. The results of this study also showed that serum SOD activity, GSH-PX activity, CAT activity, POD activity, and T-AOC significantly increased with the increase in dietary Tau, and the POD activity in the liver and jejunum mucosa increased with the increase in dietary Tau. The results of this study intimated that Tau could enhance the antioxidant activity of piglets, thereby improving the scavenging ability of ROS and RNS, and reducing the damage to the body. 

The results of this study showed that the mRNA abundance of the antioxidant enzyme genes HO-1 and GPX1 in the H-Tau group was significantly higher than in the L-Tau, M-Tau, and CON groups. The mRNA abundance of SOD1 and Nrf2 in the M-Tau and H-Tau groups was significantly higher than in the L-Tau and CON groups. The mRNA abundance of antioxidant enzyme genes SOD2 in the L-Tau, M-Tau, H-Tau groups was significantly higher than in the CON group. The results of this study intimated that Tau activates the Keap1-Nrf2/ARE antioxidant signaling pathway. Studies have shown that [15,23,24,25] hydrogen peroxide generates hypochlorous acid in the presence of chloride ions, and then hypochlorous acid reacts with Tau to produce nitrogen-chloro-Tau, which can activate Nrf2 and make it enter the nucleus to interact with the antioxidant reaction element (ARE), up-regulating the transcriptional levels of HO-1, CAT, SOD, and GSH-Px [20]. Tau not only enhances the scavenging ability of ROS and RNS but also reduces its production efficiency. It has been found [26,27,28] that Tau can enhance the synthesis efficiency of the electron transport chain-related proteins by enhancing the precise location of codons and anti-codons, such as NADH dehydrogenase subunit 5 (ND5), NADH dehydrogenase subunit 6(ND6), and cytochrome b (Cytb), which contribute to maintaining the structural and functional integrity of the electron transport chain, so as to avoid electron leakage during the transport process, and thus greatly reduce the generation of ROS. 

The results of this study also showed that serum GSSG concentration significantly decreases with the increase in dietary Tau and serum GSH concentration and the ratio of GSH to GSSG significantly increases with the increase in dietary Tau, indicating that Tau increased the reducing capacity of glutathione and weakened its oxidation capacity, and indirectly reflected that Tau could improve the antioxidant potential of glutathione. Tau can maintain the redox ability of glutathione by regulating the pH in mitochondria [16]. 

The results of this study also showed that the GSHS activity in the liver and jejunum mucosa was significantly increased with the increase in dietary Tau, indicating that Tau improved the activity of glutathione synthase in the liver and jejunum mucosa, possibly because Tau and glutathione share the same precursor cysteine [17]. Exogenous Tau supplementation can reduce the endogenous synthesis of Tau by hepatocytes, thus limiting the participation of cysteine in the pathway of Tau synthesis. This will save a lot of precursor cysteine for the synthesis of glutathione, and a lot of saved cysteine activates glutathione synthase as a substrate and then synthesizes more glutathione, increasing the antioxidant capacity of the body.

Studies have shown that oxidative stress caused by weaning can cause the destruction of the jejunal villi, accompanied by lamina propria edema, significantly reduce the height and width of the jejunal villi, and significantly increase the CD [21]. These changes will seriously reduce the digestion and absorption function and then seriously slow down the growth and development speed of piglets. In this study, Tau was found to significantly improve VH and the ratio of VH to CD in the jejunum and ileum, which was consistent with previous studies [29]. The small intestine epithelium is renewed every 2–5 days and is the most regenerative tissue in mammals [30]. The intestinal epithelium is composed of many repetitive crypt-villus units that undergo a continuous cycle of renewal and repair [31]. The crypt was the proliferative area, and the early progenitor cells produced by the intestinal stem cells (ISCs) at the bottom migrated to the crypt-villus axis and differentiated into epithelial cells [32], thereby repairing intestinal damage. 

However, the repair efficiency of ISCs for intestinal injury is closely related to the redox status. Excessive accumulation of ROS in mammals can inhibit the proliferation and differentiation of ISCs. Studies have found that the proliferation and differentiation process of ISCs is blocked under high ROS levels [33]. Further studies have confirmed that the activation of the Keap1/Nrf2 signaling pathway significantly reduces the ROS content in the intestine, creating an appropriate ROS content for intestinal cells and enhancing the activity of ISCs [34]. In addition, Keap1/Nrf2 signaling may work synergistically with Notch signaling to determine the proliferation and differentiation of ISCs. Studies have found that the functional regulatory region of the Notch1 gene is located at the proximal end of the Nrf2 promoter [35], which may be activated with the activation of Nrf2. It can negatively regulate the Notch downstream effector Math1, thus disrupting the Notch cascade and accelerating the proliferation and differentiation of mouse ISCs [36]. These results suggest that Tau can maintain the REDOX state required for the proliferation and differentiation of ISCs by activating the Keap1/Nrf2/ARE antioxidant pathway and increasing the synthetic content of glutathione, and may activate the functional regulatory region gene of Notch1 gene while activating Nrf2, thus interfering with the Notch signaling pathway, enhancing the proliferation and differentiation ability of ISCs, and improving the repair efficiency of intestinal damage.

Studies have found that oxidative stress can reduce the expression abundance of intestinal protein OCLN, CLDN-1, and ZO-1 [37] and significantly increase the level of lipopolysaccharide in blood [32]. This indicates that oxidative stress can indeed increase the permeability of intestinal mucosa and cause endotoxins to enter the blood through mucosa, further threatening the health of piglets. Dietary Tau supplementation could significantly up-regulate the mRNA abundance and content of ZO-1, OCLN, and CLDN-1 in jejunal mucosa, and significantly reduce the content of D-LA in blood in this study. This suggests that Tau can reduce intestinal mucosal permeability by restoring the tight connections between cells, thereby inhibiting enteric metabolites from crossing the intestinal defense barrier into the bloodstream, reducing the risk of enteric diseases. Previous studies found that the inhibition or deletion of Nrf2 decreased the tight junction protein expression in cells or mice [38,39]. The effect of Nrf2 on intestinal barrier function is possibly indirectly promoted in many ways, which may be associated with the regulation of Nrf2-mediated inflammation, oxidative stress, T cell activation, and autophagy [40]. In addition, Nrf2 might directly regulate tight junction protein expression, and it has been reported that Nrf2 binds to the promoter of claudin 4, which has the potential to result in an increase in claudin 4 transcription [41]. Future research is needed to determine whether Nrf2, as a transcription factor, binds the promoter of other tight junction genes to promote their transcription.

Studies have shown [42] that oxidative stress can significantly reduce the abundance of Lactobacillus and Faecalibacterium in the guts of piglets, while significantly increasing the abundance of Clostridium cluster I. Lactobacillus is a beneficial bacterium that can resist the invasion and colonization of exogenous pathogenic microorganisms [43], and Faecalibacterium is a resident anti-inflammatory probiotic in the gut of healthy animals, and Clostridium cluster I is a conditional pathogenic bacterium [44]. The imbalance of intestinal microbiota can lead to a variety of intestinal diseases [45]. Therefore, the above study found that after oxidative stress of piglets, the abundance of beneficial bacteria in the intestine is down, and the abundance of harmful bacteria is up, which indicates that oxidative stress can cause a variety of intestinal diseases by changing the microbiota in the intestine, and then adversely affect the digestion, absorption, growth, and development of piglets.

Interestingly, this appears to be reversed when Tau is added to the feed. The reason why Tau changes the microbiota of the cecum may be that it changes the microenvironment of the cecum in some way. It is reported [46] that Tau, as a sulfite donor, can produce hydrogen sulfide in the intestine, and hydrogen sulfide is an important metabolic product in the intestine, which can have both beneficial and harmful effects on the intestinal environment. Tau showed certain selectivity to the microorganisms colonized in the cecum in this way. Tau causes changes in the composition and proportion of microbiota species, although it does not change the total number of microbiota species and their quantitative distribution in the cecum. At the phylum level, the experiment results showed that Tau significantly increased the proportion of Firmicutes and significantly decreased the proportion of spirobacteroides and Proteobacteria. Although there was no significant effect on Bacteroides, the proportion of Bacteroides showed a significant downward trend with the increase in the Tau dose. According to relevant reports, Firmicutes can improve energy utilization in the diet, and Bacteroides are common microorganisms that degrade polysaccharides [47], and the ratio of Firmicutes to Bacteroides is usually positively correlated with intestinal digestion and absorption [48,49].

It was also observed in this experiment that the feed utilization rate of piglets was significantly improved after the addition of Tau in the diet, which indicates that Tau may improve the production performance of animals by improving the intestinal microbiota. Proteobacteria are a group of highly adaptable and potentially pathogenic bacteria in the gastrointestinal tract, and the imbalance of intestinal microbiota in the colon is usually accompanied by an increase in Proteobacteria abundance [50]. Spirochaeta is mostly pathogenic bacteria or conditional pathogenic bacteria, which has a potential induction effect on piglet diarrhea [51]. The proportion of Proteobacteria and Spirochaeta decreased significantly after the addition of Tau in the diet, which also indicates that Tau can limit the growth of harmful bacteria and is more conducive to intestinal health. At the generic level, Tau significantly increased the proportion of *Clostridium_sensu_stricto_1*, *Terrisporobacter,* and *Turicibacter*. *Clostridium_sensu_stricto_1*, *Terrisporobacter,* and *Turicibacter* all belong to the Firmicutes group of probiotics [52]. The study found that *Clostridium_sensu_stricto_1* can utilize mucous-derived sugars, such as glucose, and can also act as a fiber-degrading bacterium that degrades cellulose and hemicellulose [53], thereby improving the animal’s utilization capacity of coarse fiber in feed. *Terrisporter* is a lactic acid-degrading bacterium. Under anaerobic conditions, *Terrisporter* can degrade lactic acid [54], thereby avoiding acidosis and intestinal damage caused by excessive lactic acid in the intestine. Therefore, the results of this experiment also indicate that Tau may improve the intestinal nutrient utilization capacity of piglets by increasing the proportion of beneficial bacteria and then maintaining intestinal health by improving microbiota.

The above results are sufficient to prove that the addition of Tau in feed can alleviate the intestinal oxidative damage induced by early weaning of piglets. Meanwhile, we also observed that the addition of Tau in feed can significantly reduce the diarrhea index and improve the feed conversion rate. These results indicate that Tau can improve feed utilization ability by alleviating intestinal oxidative damage of piglets. In addition, we also speculate that this beneficial effect may be related to Tau improving the synthesis capacity of GSH. Studies have shown that GSH can participate in the absorption of amino acids through the *γ*-glutamyl cycle, and the extracellular amino acids react with GSH under the action of *γ*-glutamul transpeptidase (*γ*-GT) to generate *γ*-glutamyl-amino-acids and cysteinyl-glycine into the cell. In the cytoplasm, *γ*-glutamyl amino acids are catalyzed by *γ*-glutamyl cyclotransferase to decompose into amino acids and 5-hydroxyproline. 5-hydroxyproline is hydrolyzed to glutamic acid under the catalysis of 5-hydroxyprolinase, cysteine-glycine is hydrolyzed to cysteine and glycine by dipeptidase, and GSH can be synthesized in the cell [55]. GSH can also reduce Fe^3+^ to Fe^2+^, and Fe^2+^ is more easily absorbed by the intestinal mucosa, thus promoting iron absorption [56]. Studies have shown that changes in the ratio of GSSG/GSH can directly or indirectly affect the activity of enzymes related to carbohydrate metabolism [57] and the activity of Na+-dependent D-glucose carriers, and the reduction in this ratio can improve the carrying capacity of glucose [58].

This study found that the final weight, weight gain, ADG, and feed conversion rate of piglets increased with the increase in dietary Tau, while the diarrhea index of piglets decreased with the increase in dietary Tau. Similarly, some studies have found that Tau can significantly improve the growth performance of piglets fed a high plant protein diet but has no effect on piglets fed a high animal protein diet [59]. The reason for this divergence may be that almost all plant feeds do not contain Tau, while animal feeds are rich in Tau [60]. It is worth mentioning that Tau was the most abundant amino acid in red algae, especially in the *Porphyra species*. *P. tenera* and *P. haitanensis* belong to the *Porphyra species*, and they contained high levels of Tau, 975.04 mg and 645.55 mg in 100 g dry weight, respectively [61]. The Tau levels of *P. tenera* and *P. haitanensis* are about the same as fishmeal and much higher than whey powder; in addition, their crude protein contents ranged from 32.16~36.88% [60,61]. Therefore, it is of great significance to study “how to develop and utilize *P. tenera* and *P. haitanensis* as a functional protein feed in animal husbandry industry”.

## 5. Conclusions

In this study, it was found that adding the proper amount of Tau to feed can significantly alleviate intestinal oxidative damage of piglets by enhancing the antioxidant capacity of the body, and repairing intestinal damage can improve the digestion and absorption ability of piglets. In the concentration range of this experiment, the overall improvement was positively correlated with the added dose of Tau, and the improvement effect was most prominent in the H-Tau group (0.4% Tau). It is worth noting that the overall results of this study showed that the repair effect of Tau on intestinal oxidative damage and the improvement effect on feed utilization capacity were linearly correlated with the dosage of Tau. This shows that the dosage of 0.4% Tau added to the feed is not the best dose, and continuing to increase the concentration of Tau in the feed may make the alleviation effect of intestinal oxidative damage better, but the specific effect needs to be confirmed by experiment.

## Figures and Tables

**Figure 1 antioxidants-12-01852-f001:**
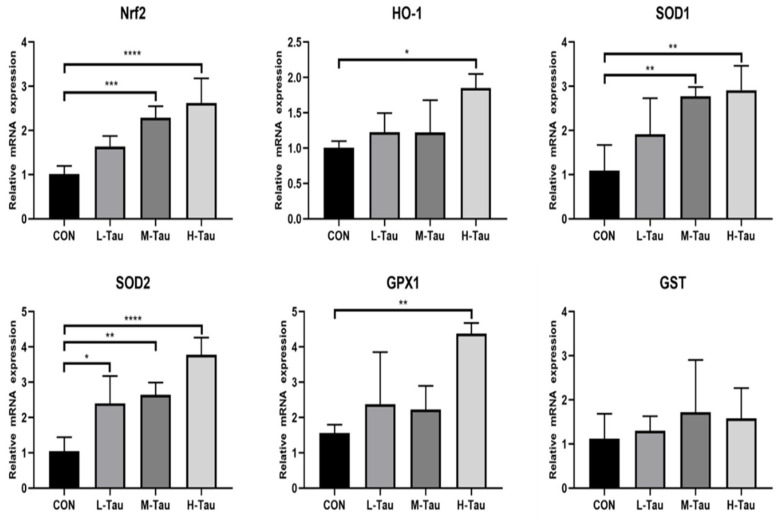
Effect of Tau on the mRNA abundances of anti-oxidase genes in the liver. (* *p* ≤ 0.05, ** *p* ≤ 0.01, *** *p* ≤ 0.001, **** *p* ≤ 0.0001).

**Figure 2 antioxidants-12-01852-f002:**
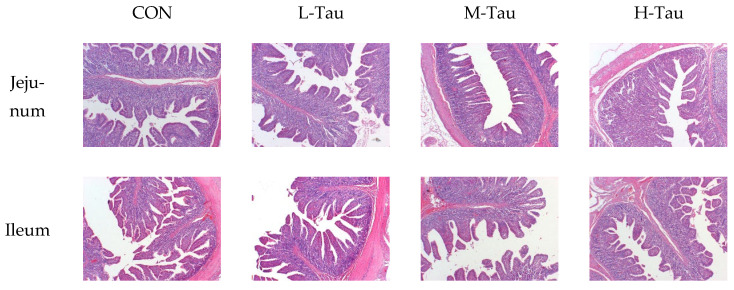
Effect of dietary taurine supplementation on intestinal tissue sections of piglets.

**Figure 3 antioxidants-12-01852-f003:**
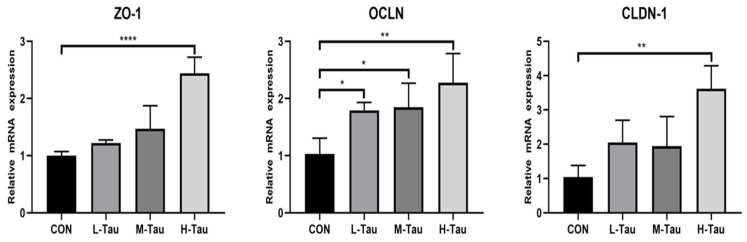
Effect of Tau on the mRNA abundances of tight junction protein genes in the liver. * *p* ≤ 0.05, ** *p* ≤ 0.01, **** *p* ≤ 0.0001.

**Figure 4 antioxidants-12-01852-f004:**
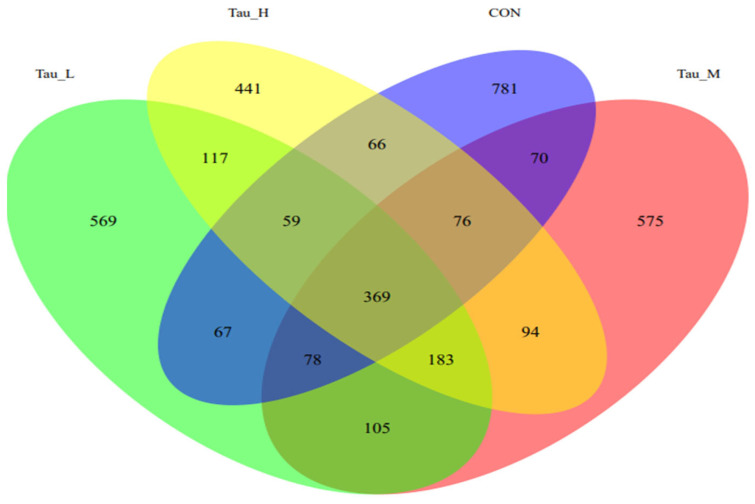
Venn diagram of microbial ASV in cecal contents of weaned piglets.

**Figure 5 antioxidants-12-01852-f005:**
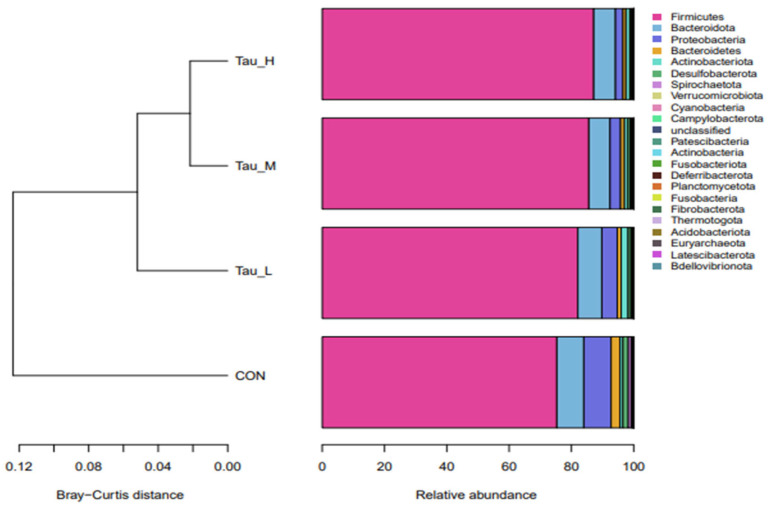
Microbial clustering map of cecal contents at the phylum level of weaned piglets.

**Figure 6 antioxidants-12-01852-f006:**
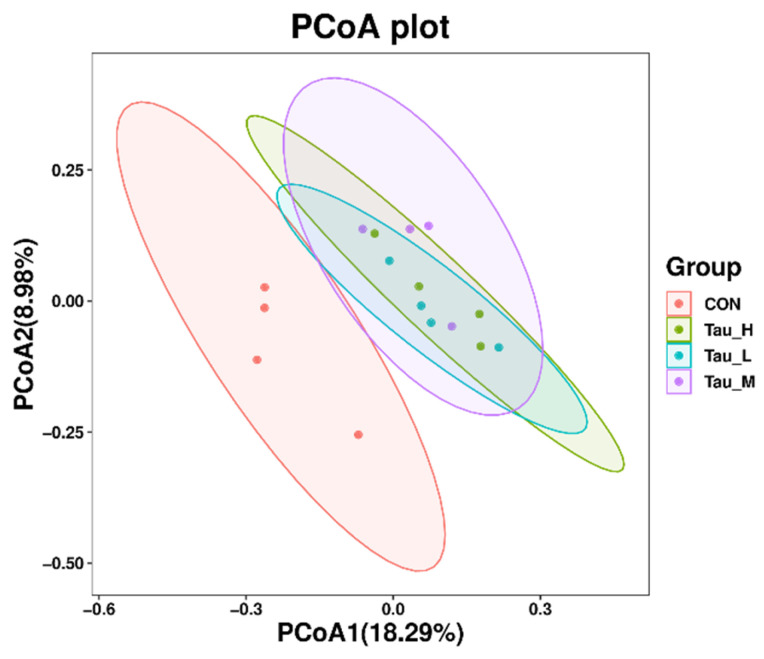
Plot of microbial PCoA analysis of cecal contents of weaned piglets.

**Table 1 antioxidants-12-01852-t001:** Ingredients and nutrient levels of basal diets.

Ingredients	Contents	Nutritional Level ^2^	Contents
Corn (%)	61.49	DE(MJ/kg)	14.34
Soybean meal (%)	12.00	CP (%)	18.42
Puffed soybean meal (%)	6.08	Ca (%)	0.78
Fish meal (%)	5.00	Total P (%)	0.67
Whey powder (%)	10.00	Available P (%)	0.40
Fatty powder (%)	1.59	Lysine (%)	1.42
Calcium hydrophosphate (%)	0.93	Methionine (%)	0.51
Mountain flour (%)	0.64	Lysine + Cystine (%)	0.78
NaCl (%)	0.30	Threonine (%)	0.90
L-lysine hydrochloride (%)	0.57	Tryptophan (%)	0.21
Methionine (%)	0.21	Taurine	0.037
Threonine (%)	0.20		
Premix ^1^ (%)	1.00		
Total (%)	100		

^1^ Provided the following per kilogram of diet: 10,500 IU VA; 3000 IU VD; 22.5 IU VE; 3.0 mg VK; 15 mg pantothenic; 7.5 mg riboflavin; 1.5 mg folic acid; 30.0 mg niacin; 3.0 mg thiamine; 4.5 mg VB6; 0.12 mg Biotin; 0.03 mg VB12; 4.0 mg Mn; 100 mg Zn; 104 mg/kg Fe; 6.0 mg Cu; 0.3 mg Se; 0.3 mg I; 600 mg sweetening agents; 800 mg choline chloride; 200 mg antioxidant. ^2^ Except for ME and Tryptophan, which are calculated values, nutrient contents are measured values.

**Table 2 antioxidants-12-01852-t002:** Primer sequences of the genes.

Gene Name	ID Number	Primer Sequence	Product Length (bp)
*GAPDH*	XM_021091114	F: CGAGATCCCGCCAACATCAA R: CCCCACCCTTCAAGTGAGC	109
*Nrf2*	MH101365.1	F: TGCAGCTTTTGGCAGAGACA R: AGGAGCAATGAAGACTGGGC	119
*ZO-1*	AJ318101.1	F: ATTCCTGAGGGAATTGGGCAG R: GCAGGGTTTCACCTTTCTCCT	94
*OCLN*	NM_001163647.2	F: ATATGGTGGAGAGACGCCCA R: AAGGGTAGCCCATACCACCT	230
*SOD2*	NM_214127.2	F: TTGCAGATTGCCGCTTGTTC R: TTACTTTTTGCAAGCGGCGT	195
*GPX1*	NM_214201.1	F: GAAGTGTGAGGTGAATGGCG R: TGCGATGTCATTGCGACACA	145
*GST*	NM_214300.2	F: CTCAAGCTTTGGCAAGGGAG R: TGGGCTCTTCGTACACGTTC	78
*HO-1*	NM_001004027.1	F: TACCGCTCCCGAATGAACAC R: GTCACGGGAGTGGAGTCTTG	209
*SOD1*	NM_001190422.1	F: AAGGCCGTGTGTGTGCTGAA R: AGTGGCCACACCATCTTTGC	279
*CLDN-1*	XM_005670262.3	F: CTAGTGATGAGGCAGATGAA R: AGATAGGTCCGAAGCAGAT	250

**Table 3 antioxidants-12-01852-t003:** Effect of dietary taurine supplementation on growth performance and the diarrhea index of piglets.

Items	Treatments				SEM	*p* Value		
	CON	L-Tau	M-Tau	H-Tau		Linear	Quadratic	Cubic
Initial weight (kg)	6.73	6.74	6.73	6.73	0.22	1.00	1.00	1.00
Final weight (kg)	12.03 ^b^	13.09 ^a^	13.22 ^a^	13.54 ^a^	0.255	0.001	0.026	0.078
Weight gain (kg)	5.81 ^b^	6.26 ^b^	6.49 ^a^	7.05 ^a^	0.156	0.001	0.028	0.058
ADG (kg/d)	166 ^b^	179 ^b^	186 ^a^	194 ^a^	4.46	0.047	0.028	0.058
ADFI (kg/d)	442	458	459	462	17.87	0.837	0.773	0.637
Feed conversion rate	0.38 ^b^	0.39 ^b^	0.40 ^a^	0.42 ^a^	0.015	0.03	0.036	0.091
Diarrhea index	1.28 ^a^	1.05 ^b^	0.87 ^bc^	0.72 ^c^	0.22	<0.001	0.573	0.973

^a, b, c^ Values with different letter superscripts within the same row mean significant difference (*p* < 0.05). ADG: average daily gain; ADFI: Average daily feed intake.

**Table 4 antioxidants-12-01852-t004:** Effect of dietary taurine supplementation on antioxidant properties of piglets.

Items	Treatments				SEM	*p* Value		
	CON	L-Tau	M-Tau	H-Tau		Linear	Quadratic	Cubic
Serum								
MDA (nmol/L)	11.2 ^a^	12.5 ^a^	9.66 ^ab^	6.97 ^b^	1.86	0.003	0.053	0.320
NO (μmoL/mL)	0.34 ^a^	0.34 ^a^	0.25 ^b^	0.24 ^b^	0.04	0.001	0.935	0.100
SOD (U/mL)	209 ^b^	197 ^b^	249 ^b^	387 ^a^	46.51	<0.001	0.007	0.809
GPX (U/mL)	671 ^b^	701 ^b^	841 ^a^	842 ^a^	64.46	0.003	0.686	0.118
CAT (U/mL)	17.40 ^b^	15.72 ^b^	24.16 ^a^	24.14 ^a^	2.57	0.001	0.546	0.009
POD (U/mL)	6.25 ^b^	6.52 ^b^	8.75 ^ab^	12.99 ^a^	2.82	0.007	0.204	0.996
T-AOC (μmol/mL)	0.12 ^c^	0.13 ^bc^	0.15 ^b^	0.19 ^a^	0.02	<0.001	0.162	0.833
O_2_^•−^ clearance rate (%)	17.35 ^b^	16.96 ^b^	17.81 ^ab^	18.64 ^a^	0.73	0.013	0.119	0.457
^•^OH clearance rate (%)	3.50 ^b^	3.54 ^b^	3.65 ^b^	3.97 ^a^	0.20	<0.001	0.187	0.787
GSH (μmol/mL)	0.09 ^ab^	0.08 ^b^	0.13 ^ab^	0.19 ^a^	0.06	0.033	0.310	0.717
GSSG (nmol/mL)	3.67 ^ab^	3.81 ^a^	3.61 ^ab^	2.71 ^b^	0.62	0.050	0.138	0.810
GSH/GSSG	25.29 ^b^	22.71 ^b^	36.75 ^ab^	80.79 ^a^	30.52	0.019	0.145	0.845
Liver								
O_2_^•−^ clearance rate (%)	17.37 ^b^	16.99 ^b^	18.21 ^a^	18.24 ^a^	0.42	0.004	0.396	0.021
^•^OH clearance rate (%)	3.38 ^b^	3.64 ^ab^	3.79 ^ab^	4.00 ^a^	0.23	0.005	0.867	0.763
POD (U/mg prot)	4.71 ^ab^	3.93 ^b^	5.62 ^ab^	6.62 ^a^	1.41	0.040	0.252	0.372
GSHS (U/g)	0.42 ^c^	0.39 ^c^	0.49 ^b^	0.58 ^a^	0.04	<0.001	0.013	0.099
Jejunal mucosa								
POD (U/mg prot)	2.91 ^b^	2.94 ^b^	4.79 ^ab^	6.33 ^a^	1.76	0.011	0.429	0.621
O_2_^•−^ clearance rate (%)	16.37 ^b^	15.88 ^b^	16.63 ^b^	18.03 ^a^	0.45	<0.001	0.002	0.572
^•^OH clearance rate (%)	3.36 ^b^	3.48 ^b^	3.91 ^a^	4.20 ^a^	0.20	<0.001	0.413	0.366
GSHS (U/g)	0.38 ^b^	0.34 ^b^	0.41 ^b^	0.69 ^a^	0.05	<0.001	<0.001	0.423

^a, b, c^ Values with different letter superscripts within the same row mean significant difference (*p* < 0.05). SOD: superoxide dismutase; GPX: glutathione peroxidase; CAT: catalase; POD: peroxidase; T-AOC: total antioxidant capacity; GSH: glutathione; GSSG: oxidized glutathione; MDA: malondialdehyde; NO: nitric oxide; GSHS: glutathione synthase.

**Table 5 antioxidants-12-01852-t005:** Effect of dietary taurine supplementation on intestinal morphology and structure of piglets.

Items	Treatments				SEM	*p* Value		
	CON	L-Tau	M-Tau	H-Tau		Linear	Quadratic	Cubic
Jejunum								
Weight (g)	277.01	309.64	279.62	335.56	53.33	0.245	0.670	0.237
Weight index (g/kg)	26.57	28.17	25.17	26.62	6.11	0.838	0.981	0.521
Length (cm)	744.50	785.00	787.50	857.50	132.48	0.271	0.828	0.728
Length index (cm/kg)	71.22	70.57	71.05	68.09	14.27	0.785	0.874	0.888
VH (μm)	235.95 ^c^	286.75 ^bc^	320.24 ^ab^	368.00 ^a^	47.55	0.003	0.952	0.774
CD (μm)	82.11	74.37	73.00	71.80	18.38	0.447	0.728	0.883
VH/CD	3.46 ^b^	3.98 ^b^	4.65 ^ab^	6.17 ^a^	1.19	0.011	0.432	0.800
Ileum								
Weight (g)	117.5	155.11	149.28	127.01	45.45	0.839	0.232	0.797
Weight index (g/kg)	11.03	13.98	12.87	10.03	3.67	0.649	0.159	0.783
Length (cm)	238.25	366.25	373.00	287.50	104.61	0.521	0.064	0.903
Length index (cm/kg)	22.79	32.77	32.49	22.68	8.34	0.974	0.035	0.970
VH (μm)	286.69 ^b^	301.40 ^b^	335.97 ^ab^	384.53 ^a^	46.13	0.019	0.512	0.957
CD (μm)	106.96	91.19	80.22	73.04	25.13	0.068	0.739	0.986
VH/CD	2.92 ^b^	3.56 ^b^	4.61 ^ab^	5.81 ^a^	1.13	0.004	0.650	0.915

^a, b, c^ Values with different letter superscripts within the same row mean significant difference (*p* < 0.05). VH: villus height; CD: crypt depth.

**Table 6 antioxidants-12-01852-t006:** Effect of dietary taurine supplementation on the intestinal mucosal barrier of piglets.

Items	Treatments				SEM	*p* Value		
	CON	L-Tau	M-Tau	H-Tau		Linear	Quadratic	Cubic
Serum								
D-LA (μmol/L)	122.51 ^a^	107.10 ^a^	94.74 ^ab^	76.32 ^b^	17.98	0.003	0.870	0.825
Jejunal mucosa								
CLDN-1 (ng/g)	586.88 ^c^	632.24 ^c^	1017.64 ^b^	1469.29 ^a^	126.29	<.001	0.007	0.351
ZO-1 (ng/g)	756.72 ^b^	714.90 ^b^	879.10 ^b^	1291.94 ^a^	133.31	<0.001	0.007	0.890
OCLN (ng/g)	39.78 ^c^	32.09 ^c^	66.02 ^b^	136.85 ^a^	9.54	<0.001	<0.001	0.841

^a, b, c^ Values with different letter superscripts within the same row mean significant difference (*p* < 0.05). D-LA: D-lactose; CLDN-1: claudin 1; ZO-1: zonula occluden 1; OCLN: occludin.

**Table 7 antioxidants-12-01852-t007:** Effect of dietary taurine supplementation on the cecal microbiota diversity of piglets.

Items	Treatments				SEM	*p* Value		
	CON	L-Tau	M-Tau	H-Tau		Linear	Quadratic	Cubic
Alpha diversity in								
Observed_otus	594.00	632.25	610.00	554.50	106.52	0.566	0.396	0.911
Shannon	7.19	6.73	6.30	6.35	0.77	0.111	0.520	0.788
Simpson	0.97	0.95	0.92	0.93	0.04	0.202	0.507	0.528
Chao1	595.23	641.05	618.02	555.85	110.07	0.577	0.346	0.906
Pielou_e	0.78	0.72	0.68	0.70	0.07	0.084	0.319	0.775
Goods_coverage	1.00	1.00	1.00	1.00	0.00	0.809	0.032	0.872
Phylum level in								
Firmicutes	75.33 ^b^	82.06 ^a^	85.60 ^a^	87.17 ^a^	4.17	0.001	0.240	0.898
Proteobacteria	10.48 ^a^	5.88 ^ab^	4.25 ^b^	2.84 ^b^	3.23	0.005	0.342	0.710
Bacteroidetes	11.49	8.94	8.03	7.94	4.75	0.297	0.614	0.939
Actinobacteriota	0.94	2.23	1.06	1.26	0.82	0.914	0.216	0.060
Spirochaetota	0.95 ^a^	0.18 ^b^	0.22 ^b^	0.11 ^b^	0.33	0.006	0.07	0.233
Verrucomicrobiota	0.23	0.41	0.21	0.34	0.38	0.899	0.891	0.418
Cyanobacteria	0.09	0.11	0.52	0.16	0.28	0.340	0.191	0.085
Generic level								
Clostridium_sensu _stricto_1	5.25 ^b^	24.10 ^a^	37.08 ^a^	34.71 ^a^	11.55	0.002	0.091	0.721
Streptococcus	13.49	5.92	2.72	5.42	6.89	0.101	0.162	0.922
Lachnospiraceae	6.08	5.96	4.56	2.65	4.48	0.266	0.696	0.938
Terrisporobacter	0.56 ^b^	4.19 ^a^	4.74 ^a^	3.12 ^ab^	1.93	0.081	0.019	0.836
UCG-005	5.40 ^a^	1.47 ^b^	2.39 ^b^	2.29 ^b^	1.32	0.015	0.014	0.072
T34_unclassified	4.93	2.38	2.25	1.08	2.42	0.053	0.579	0.533
Agathobacter	3.33	2.53	2.44	1.62	1.69	0.190	0.990	0.707
Turicibacter	0.42 ^b^	1.93 ^ab^	3.59 ^a^	3.39 ^a^	1.45	0.007	0.263	0.543
Ruminococcus	3.13	1.37	1.75	1.87	1.22	0.234	0.146	0.395
Clostridium_sensu_stricto_2	2.15	1.57	1.81	2.51	1.61	0.716	0.440	0.918
Streptococcus	0.53	2.52	1.74	1.78	1.45	0.380	0.203	0.292

^a, b^ Values with different letter superscripts within the same row mean significant difference (*p* < 0.05).

## Data Availability

The 16S rDNA sequencing raw data are available at NCBI under the accession number PRJNA974223 by web link (https://www.ncbi.nlm.nih.gov/bioproject/PRJNA974223 (accessed on 19 May 2023)). The rest of the raw data supporting the conclusions of this article will be made available by the authors without undue reservation.

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
