# Peer review of "Dietary Taurine Improves Growth Performance and Intestine Health via the GSH/GSSG Antioxidant System and Nrf2/ARE Signaling Pathway in Weaned Piglets"

_antioxidants, 2023, doi:10.3390/antiox12101852_

Round 1
Reviewer 1 Report
The manuscripts described a feeding experiment in piglets, aiming to assess the beneficial effects of a supplementation of the diet with increasing amounts (3 concentrations tested) of the amino acid taurine. The paper was driven by the hypothesis that Taurine improves growth performance and intestine healthy via the GSH/GSSG antioxidant system. The parameters studied included next to the markers of oxidative stress predominantly biomarkers of gut health, included a 16s rRNA microbiota analysis. The obtained results convincingly and dose-dependently demonstrate the added value of feed supplementation with Taurine.
The manuscript is well written and the selection and description (M and M and results) of the analysed parameters is complete and according to the state of the art. There are only very few, mainly editorial remarks that would require the attention of the authors:
Line 18-19: studied the effect of…..
Line 21: keep it simple: 48 piglets were assigned to 4 groups of 12 individuals (also modify lines 102-103).
Line 57: replace “flora” by microflora or microbiota (in line with the other parts of the manuscript; consider that microbiota is currently the most accepted wording).
Table 1: this table is partly redundant (alle groups are the same, hence 1 column would be sufficient). The definition of the concentrations (given as dilutions) of Taurine in the table are misleading and needs to be corrected. The correct description is given in the subscript to the table (footnote 2). Did the authors conduct any analytical control measures, i.e. measuring the actual taurine content in the 4 individual diets?
Lines 126 and 135: The wording “Index Determination” (certainly for ELISA measurements) is very unusual in this context. Consider replacing this by “biochemical parameters of oxidative stress” and “indicators of intestinal barrier integrity” or something similar.
Line 172 and Table 4: Please describe how the “Clearance rate of O and OH radicals” was calculated (not clearly described in section 2.3.).
Table 8: Consider presenting the basic in-life parameters such as ADWG and FCR at the beginning of the result section together with the clinical diarrhoea scores.
Lin 296: please rephrase: … share the same “synthetic raw material”…: by “share the same precursor” or something similar. Similarly line 298: this will save a lot of cysteine raw material … should be rephrased.
Line 452-454: the authors might want to consider / discuss in this context that algae contain also quite high amounts of Taurine and are increasingly used as feed additives.
Minor editing of English language required
Author Response
Section A: "Responses to the Comments by reviewer 1"
General comments: The manuscripts described a feeding experiment in piglets, aiming to assess the beneficial effects of a supplementation of the diet with increasing amounts (3 concentrations tested) of the amino acid taurine. The paper was driven by the hypothesis that Taurine improves growth performance and intestine healthy via the GSH/GSSG antioxidant system. The parameters studied included next to the markers of oxidative stress predominantly biomarkers of gut health, included a 16s rRNA microbiota analysis. The obtained results convincingly and dose-dependently demonstrate the added value of feed supplementation with Taurine.The manuscript is well written and the selection and description (M and M and results) of the analysed parameters is complete and according to the state of the art. There are only very few, mainly editorial remarks that would require the attention of the authors:
Authors' responses: Thank reviewer for valuable comments and suggestions. We have made comprehensive modifications according to the suggestions. The detailed modification were presented in revised manuscript (highlighted in yellow).
Comment 1: Line 18-19: studied the effect of…..
Authors' responses: thanks. I have realized the grammatical error of this sentence, and I've rewritten that sentence. (Please see lines 18-19 in the revised version, highlighted in yellow).
Comment 2: Line 21: keep it simple: 48 piglets were assigned to 4 groups of 12 individuals (also modify lines 102-103).
Authors' response: According to your suggestion, I have modified " A total of 48 weaned piglets (6.71 ± 0.73 kg) of 28-day-old crossbred "Duroc × Landrace × Yorkshire" were assigned to 4 groups with 12 replicates per group and 1 pig per replicate " in the original text to "48 weaned piglets (6.71 ± 0.73 kg) of 28-day-old crossbred "Duroc × Landrace × Yorkshire" were assigned to 4 groups of 12 individuals ". (Please see lines 19-20 and 89-90 in the revised version, highlighted in yellow).
Comment 3: Line 57: replace “flora” by microflora or microbiota (in line with the other parts of the manuscript; consider that microbiota is currently the most accepted wording).
Authors' responses:Thanks. Owing to your suggestion, I have replaced all the "flora" in the first draft with "microflora". (Please see line 58, 382, 403,405 and 421 in the revised version, highlighted in yellow).
Comment 4: Table 1: this table is partly redundant (alle groups are the same, hence 1 column would be sufficient). The definition of the concentrations (given as dilutions) of Taurine in the table are misleading and needs to be corrected. The correct description is given in the subscript to the table (footnote 2). Did the authors conduct any analytical control measures, i.e. measuring the actual taurine content in the 4 individual diets?
Authors' responses: Thank you for your good suggestions. We had changed Table1 and only provide a basal diet and 0.037% Taurine in basal diet.
Comment 5: Lines 126 and 135: The wording “Index Determination” (certainly for ELISA measurements) is very unusual in this context. Consider replacing this by “biochemical parameters of oxidative stress” and “indicators of intestinal barrier integrity” or something similar.
Authors' responses: Thanks. Owing to your suggestion, I have combined the two parts into one and titled it "Biochemical parameters of oxidative stress and indicators of intestinal barrier integrity". (Please see line 126 in the revised version, highlighted in yellow).
Comment 6: Line 172 and Table 4: Please describe how the “Clearance rate of O and OH radicals” was calculated (not clearly described in section 2.3.).
Authors' responses: thanks, we had described how the “Clearance rate of O and OH radicals” was calculated in context. (Please see section 2.4. highlighted in yellow, in revised manuscript)
- OH clearance rate: •OH was formed by Fenton reaction between H2O2 and Fe2+. •OH reacted with C₇H₆O₃, and the product showed special absorption at 510nm. Adding different samples to the system with the same concentration of •OH, after 20 minutes of reaction, the active substance in the sample can remove part of •OH, and then reduce the absorption value at 510nm, according to "•OH clearance (%)=[A blank -(A determination -A control)]÷A blank ×100%",the larger the percentage value, the stronger the scavenging ability of the sample to •OH. You can find the detailed instructions for the kit at the Link(http://m.ruixinbio.com/product/detail/35.html)
O2•- clearance rate: O2•- reacts with the reduced material to produce purplish red compound, which has a characteristic absorption peak at 570nm, and the removal ability of the sample to O2•- is negatively correlated with the absorption value at 570nm. According to O2•- clearance I%=[1-(A determination -A control)÷A blank]×100%, the larger the percentage value, the stronger the scavenging ability of the sample to O2•-. You can find the detailed instructions for the kit at the Link(http://m.ruixinbio.com/product/detail/43.html)
Comment 7: Table 8: Consider presenting the basic in-life parameters such as ADWG and FCR at the beginning of the result section together with the clinical diarrhoea scores.
Authors' responses: Thanks, we had presented the basic in-life parameters such as ADWG and FCR at the beginning of the result section together with the clinical diarrhoea scores. (Please see “the result section”. highlighted in yellow, in revised manuscript).
Comment 8: Lin 296: please rephrase: … share the same “synthetic raw material”…: by “share the same precursor” or something similar. Similarly line 298: this will save a lot of cysteine raw material … should be rephrased.
Authors' responses: Thanks. Owing to your suggestion, I have changed "raw material" to "precursor". (Please see line 318 and 320 in the revised version, highlighted in yellow).
Comment 9: Line 452-454: the authors might want to consider / discuss in this context that algae contain also quite high amounts of Taurine and are increasingly used as feed additives.
Authors' responses: Thank you for your good suggestion. We had added text content about “algae contain also quite high amounts of Taurine and are increasingly used as feed additives” in “discuss in this context”. (Please see lines 448-454 in the discussion version, highlighted in yellow).
Comment 10: Comments on the Quality of English Language. Minor editing of English language required.
Authors' response: Thanks. Based on the Authors are not from a native-English nation, We Entrusted to Editage (www.editage.cn) for English language editing and corrected many mistakes and completely revised the English language.
Section B: Additional self- modification and improvement of manuscription
- We re-write this part “the material and methods”.
- In order to ensure the accuracy of the quoted content, we carefully checked the references of the article one by one.

Reviewer 2 Report
Please check all the abbreviations carefully. For example, Taurine was defined as Tau but there are too many “Taurine” in the manuscript. please revise them.
Introduction
There are many running sentences that need to be divided to two senetnces for clarity.
50: The term "piglets stress response" is a bit vague. Do you mean physiological stress responses?
53-56: You might consider breaking up the long sentence starting from line 54 into two for easier comprehension.
59-66: The discussion about zinc and copper in feed is interesting, but it could be made clearer why it is relevant to this study on Taurine. The connection is made later, but it might be clearer if it was laid out more directly here.
72-74: It might be clearer if you specified what you mean by "body regulator". In what sense is Taurine a body regulator?
85-93: This section discusses the value of your research well. However, the sentence structure is a bit complex. Consider simplifying it for improved readability.
Results
Line 182: The semicolon after "SOD2 (P < 0.05);" seems misplaced.
Line 184-186: Be careful not to make too many generalizations in the results section, as these might better belong in the discussion section. For instance, "in general, the improvement effect of H-Tau group on the antioxidant capacity of the body was greater than M-Tau group" is more of a conclusion than a result.
Line 201: I'd recommend being more specific about what "clearer" means in this context. Does it mean they were more visible, more well-defined, etc.?
Line 249: I'd recommend rewording "a certain effect" to specify what kind of effect was observed.
Line 259: Similar to my earlier comment, ensure that all abbreviations are defined at first mention within the section, such as "Tau."
The discussion section is too long and many of the redundant sentences have to be omitted.
Dear Editor,
After thorough review, the manuscript appears to be well-written and insightful, but could benefit from some revisions for enhanced clarity and conciseness. My specific recommendations for revision, which include clarification of terms, improvement of sentence structure, review of abbreviations, and omission of redundant sentences, especially in the discussion section, have been detailed to the authors. With these revisions, I believe the manuscript has the potential to make a valuable contribution to the field.
Best regards,
Hosseindoust
Author Response
Section A: "Responses to the Comments by reviewer 2"
General comments: After thorough review, the manuscript appears to be well-written and insightful, but could benefit from some revisions for enhanced clarity and conciseness. My specific recommendations for revision, which include clarification of terms, improvement of sentence structure, review of abbreviations, and omission of redundant sentences, especially in the discussion section, have been detailed to the authors. With these revisions, I believe the manuscript has the potential to make a valuable contribution to the field.
Authors' responses: Thank reviewer for valuable comments and suggestions. We have made comprehensive modifications according to the suggestions. The detailed modification were presented in revised manuscript (highlighted in light blue).
Comment 1: Please check all the abbreviations carefully. For example, Taurine was defined as Tau but there are too many “Taurine” in the manuscript. please revise them.
Authors' responses: I have carefully reviewed and checked all the abbreviations carefully in manuscript, then correct all abbreviations and replaced "Taurine" with "Tau". (Please see the full text in the revised version, highlighted in light blue)
Comment 2: (Introduction) There are many running sentences that need to be divided to two senetnces for clarity.
Authors' response: In order to make the language of the article easier to understand, I made some changes for those sentences. (Please see the “introduction” section in the revised version, highlighted in light blue)
Comment 3: (line 50) The term "piglets stress response" is a bit vague. Do you mean physiological stress responses?
Authors' responses: Yes, the "piglets stress response" in my article refers to the physiological stress responses induced by early weaning of piglets, we correct it. (Please see the line 53 in the revised version, highlighted in light blue)
Comment 4: (lines 53-56) You might consider breaking up the long sentence starting from line 54 into two for easier comprehension.
Authors' responses: I have taken your valuable comments and revised the corresponding sentences. (Please see the lines 53-59 in the revised version, highlighted in light blue)
Comment 5: (lines 59-66) The discussion about zinc and copper in feed is interesting, but it could be made clearer why it is relevant to this study on Taurine. The connection is made later, but it might be clearer if it was laid out more directly here.
Authors' responses: I have carefully considered the issues you mentioned, and I think the discussion on "zinc" and "copper" in the "introduction" of the article is not very relevant to this study, so I have decided to delete this part from this manuscript.
Comment 6: (lines 72-74) It might be clearer if you specified what you mean by "body regulator". In what sense is Taurine a body regulator?
Authors' responses: we had changed it. (Please see the line 64 in the revised version, highlighted in light blue)
Comment 7: (lines 85-93) This section discusses the value of your research well. However, the sentence structure is a bit complex. Consider simplifying it for improved readability.
Authors' responses: Thanks for your advice, I have highly simplified the corresponding content. (Please see the lines 76-78 in the revised version, highlighted in light blue)
Comment 8: (Line 182) The semicolon after "SOD2 (P < 0.05);" seems misplaced.
Authors' responses: I admit that there was an error here and I have rewritten it. (Please see the lines 208-210 in the revised version, highlighted in light blue)
Comment 9: (Lines 184-186) Be careful not to make too many generalizations in the results section, as these might better belong in the discussion section. For instance, "in general, the improvement effect of H-Tau group on the antioxidant capacity of the body was greater than M-Tau group" is more of a conclusion than a result.
Authors' responses: I admit that the content you mentioned in the "results" section is indeed inappropriate, I have removed it.
Comment 10: (Line 201) I'd recommend being more specific about what "clearer" means in this context. Does it mean they were more visible, more well-defined, etc.?
Authors' response: This "clear" description is that the intestinal microenvironment does not contain damaged villi, indicating a low degree of intestinal villi damage. For clearer purposes, I have replaced "clearer" with "sharpness". (Please see the line 223 in the revised version, highlighted in light blue)
Comment 11: (Line 249) I'd recommend rewording "a certain effect" to specify what kind of effect was observed.
Authors' responses: "Microbial beta diversity" has been described in detail above in this paragraph (lines 247-255 in the revised version), and "a certain effect" here is only a brief summary.
Comment 12: (Line 259) Similar to my earlier comment, ensure that all abbreviations are defined at first mention within the section, such as "Tau."
Authors' responses: Thank you for your suggestion, I have corrected similar problems. (Please see the full text in the revised version, highlighted in light blue)
Comment 13: The discussion section is too long and many of the redundant sentences have to be omitted.
Authors' responses: I also found that my discussion section was too long and I have reduced it by about 600 words in order to make my discussion more concise. (Please see the discussion section in the revised version)
Section B: Additional self- modification and improvement of manuscription
- We re-write this part “the material and methods”.
- In order to ensure the accuracy of the quoted content, we carefully checked the references of the article one by one.
